# Scoring Systems for Clinical Colon Capsule Endoscopy—All You Need to Know

**DOI:** 10.3390/jcm10112372

**Published:** 2021-05-28

**Authors:** Trevor Tabone, Anastasios Koulaouzidis, Pierre Ellul

**Affiliations:** 1Gastroenterology Department, Mater Dei Hospital, MSD 2090 Msida, Malta; pierre.ellul@gov.mt; 2Department of Social Medicine & Public Health, Pomeranian Medical University, 70-204 Szczecin, Poland; akoulaouzidis@hotmail.com

**Keywords:** colon capsule endoscopy, colonoscopy, endoscopy score, inter-observer agreement, polyps, bowel preparation, inflammatory bowel disease, minimally invasive

## Abstract

In the constantly developing era of minimal diagnostic invasiveness, the role of colon capsule endoscopy in colonic examination is being increasingly recognised, especially in the context of curtailed endoscopy services due to the COVID-19 pandemic. It is a safe diagnostic tool with low adverse event rates. As with other endoscopic modalities, various colon capsule endoscopy scores allow the standardisation of reporting and reproducibility. As bowel cleanliness affects CCE’s diagnostic yield, a few operator-dependent scores (Leighton–Rex and CC-CLEAR scores) and a computer-dependent score (CAC score) have been developed to grade bowel cleanliness objectively. CCE can be used to monitor IBD mucosal disease activity through the UCEIS and the panenteric CECDAIic score for UC and CD, respectively. CCE may also have a role in CRC screening, given similar sensitivity and specificity rates to conventional colonoscopy to detect colonic polyps ≥ 10 mm and CRC. Given CCE’s diagnostic yield and reproducible clinical scores with high inter-observer agreements, CCE is fast becoming a suitable alternative to conventional colonoscopy in specific patient populations.

## 1. Introduction

The disruption brought about by the 2019 novel coronavirus (COVID-19) pandemic has led to a forward push with genuine innovation to expand healthcare services. Adjusting to curtailed nonemergency endoscopy services has led to a renewed interest in alternative modalities for colonic exploration, in an attempt to mitigate potential diagnostic delays. One such strategy to visualize the colon is by means of colon capsule endoscopy (CCE), which allows for a pain-free colonic assessment by eliminating the need for instrument insertion, gas insufflation or sedation [1,2]. This has allowed for the extension of CCE’s role in colorectal cancer (CRC) screening in the average-risk population, as well as an alternative for patients who refuse optical colonoscopy (OC) or in whom the latter is contraindicated [3,4,5]. Although CT colonography (CTC), like CCE, can visualise the colon without biopsy or therapy capabilities, in CTC, gas insufflation is required while concerns remain as to the ”allowed” frequency of use due to radiation exposure.

In daily endoscopic practice, the use of various scores and scoring systems allow for the standardisation of reporting to increase objectivity, the reproducibility of findings and inter-observer agreement such as a description of lesions, mucosal disease activity and the adequacy of bowel preparation [6]. Eventually, the standardisation of reporting enables a higher quality of care and facilitates decision making and the continuity of care. Given the increasing use of CCE in clinical practice, we review the currently available scoring systems used in clinical CCE.

## 2. Bowel Cleanliness Scoring

The Boston Bowel Preparation Scale (BBPS) is a visual scale in which conventional colonoscopy images are rated according to a quantitative grading based on the aggregate of separate scores obtained from the right, transverse and left colon, respectively [7]. Each colonic segment is given a score from 0 to 3, with 0 denoting an unprepared colon segment in which the mucosa could not be visualised, due to solid stool which could not be cleared. A score of 1 denotes that a portion of the colonic mucosa could be seen; however, other areas were inadequately visualised because of residual stool, staining and/or opaque liquid. A score of 2 represents an adequately visualised colonic mucosa despite a minor residual staining, small stool fragments and/or opaque liquid. When the entire colonic mucosa within a segment is well visualised with no residual staining, small stool fragments or opaque liquid, a score of 3 is given. With an Intraclass Correlation Coefficient (ICC) of 0.91, the BBPS is regarded as having a significantly high inter-rater reliability. The BBPS also demonstrates that increasingly adequate mucosal visualisation is associated with improved polyp detection rates. BBPS segment scores of 2 and 3 versus 0 and 1 were correlated with higher polyp detection rates in the left colon (OR 2.58, 95% CI 1.34–4.98) and right colon (OR 1.60, 95% CI 1.01–2.55) [7]. 

Validated scores for OC cannot be used for CCE [8]. This is because scores such as the BBPS are typically applied during the withdrawal phase of OC after the endoscopist has attempted all cleaning manoeuvres, thus reflecting the actual practice of OC. Owing to the inherent inability to perform such manoeuvres during CCE, such scores cannot be translated to CCE. Van Gossum et al., in a multicentre study comparing CCE (Pillcam^®^COLON-1, Given Imaging Ltd, Yokneam, Israel) and OC, demonstrated an association between the cleanliness level and the accuracy of CCE for detecting polyps. In terms of CCE and detection of colorectal polyps and cancer, the sensitivity of CCE was significantly higher in patients with good or excellent colon cleanliness than in those with fair or poor colon cleanliness. The latter was determined using a four-point grading scale (Table 1), a global colonic assessment, as opposed to the segmental scoring approach present in the BBPS.

In the detection of polyps ≥ 6 mm, CCE demonstrated a specificity of 75% (95% CI 65–83) in patients with good/excellent bowel cleanliness compared to 42% (95% CI 28–56) in patients with fair/poor bowel cleanliness. Similarly, for advanced adenomas (defined as adenomas ≥ 10 mm or adenomas with villous features or high-grade dysplasia), an improved sensitivity of 88% (95% CI 74–95) was noted in those with better bowel cleanliness as opposed to a sensitivity of 44% (95% CI 25–64) for those with a poorer bowel cleanliness [9]. 

### 2.1. Colon Capsule Bowel Cleansing Scores

The diagnostic yield for CCE depends on two significant factors, including bowel preparation and image interpretation [10]. Similar to OC, CCE views allow for a detailed mucosal assessment. Figure 1 demonstrates a normal colonic mucosal appearance on CCE. Various pathologies apart from those mentioned in the scoring review can be detected, such as colonic angiodysplasias (Figure 2), internal haemorrhoids (Figure 3), and diverticula (Figure 4). 

The bowel preparation regimen typically used in CCE involves large-volume (3–4 L) lavage with polyethylene glycol (PEG) in combination with prokinetics and boosters [10]. Whilst cleaning the colon, this regimen promotes capsule propulsion and creates a watery medium for the capsule to move. The relative complexity and duration of bowel preparation contrast with other noninvasive diagnostic modalities such as CTC. The latter can be performed with laxative-free regimes, which make exclusive use of faecal tagging solutions [11]. This is advantageous over the more aggressive bowel preparation regime required for CCE as it better tolerated by the elderly and a has more favourable side-effect profile in patients with renal or liver impairment [12,13]. 

#### 2.1.1. Operator-Dependent Scores

The Leighton–Rex scale is a qualitative scale that divides the colon into five segments (caecum, right colon, transverse, left colon and rectum). The five segments are graded individually with regard to overall cleansing level and bubbles grading (Table 2). This score analyses the entire video but is entirely subjective. This study recruited 40 patients, made use of the Pillcam^®^COLON-2 (Medtronic, Dublin, Ireland), and was not designed to validate the clinical utility of the tested scale [14].

Another operator-dependent bowel cleansing score is the Colon Capsule CLEansing Assessment and Report (CC-CLEAR). This is a quantitative scale where, in the study, 58 patients were recruited. The study used Pillcam^®^COLON-2. In this score, the colon was divided into three segments: right, transverse and left colon. Each segment was scored according to an estimation of the percentage of visualized mucosa (0: <50%; 1: 50–75%; 2: >75%; 3: >90%). The overall cleansing classification was a sum of each segment score, grading between excellent (8–9), good (6–7) and inadequate (0–5). Any segment scoring ≤1 resulted in an inadequate overall classification. The colon videos were reviewed by two, blinded to each other, operators and were scored by the two available operator-dependent scores: CC-CLEAR and the Leighton–Rex score scale. The CC-CLEAR interobserver agreement was superior to the Leighton–Rex scale (Kendall’s W 0.911 vs. 0.806, *p* < 0.01) [14].

One explanation for this might be due to the subjective clinical judgment on which the Leighton–Rex scale relies instead of the more objective, quantitative assessment of the percentage of visualized mucosa needed for CC-CLEAR. Moreover, the Leighton–Rex scale is more cumbersome to use in clinical practice as the colon needs to be divided into five different segments, which is a challenging task in practice, as well as the separate analysis of the “bubbles effect”, which is likely unnecessary as they equally interfere with mucosal visualization together with faecal residue. Endoscopists might find the CC-CLEAR a more user-friendly score as the colon needs to be divided into just three segments: right colon, transverse and left colon, based on the hepatic and splenic flexure landmarks.

#### 2.1.2. Computer-Dependent Scores

An automated score, similar to that of Van Weyenberg et al. for the small bowel (SB), was developed: the colonic computed assessment of cleansing (CAC) score. This was based on a still frame colorimetric analysis of the tissue colour bar. The Pillcam^®^COLON-2 system was used. The CAC score, defined as the ratio of colour intensities, red over green (R/G ratio) and red over brown (R/(R + G) ratio), was calculated for extracted colonic frames. A threshold was applied and, thus, one was able to discriminate “adequately cleansed” from “inadequately cleansed” CCE still frames. Though automated systems can apply a rapid and reproducible score, the current limitations, bubbles, luminosity and contrast, which contribute to clarity, are not considered. Furthermore, as opposed to an entire video, still frames were used, thus leading to a lot of unanalysed frames. However, such a score may circumvent the subjectivity of qualitative grading systems [15].

## 3. Mucosal Disease Activity in Inflammatory Bowel Disease

CCE is an effective tool to noninvasively assess mucosal disease activity in inflammatory bowel disease (IBD), enabling the monitoring of response to treatment, determine mucosal healing, and surveillance. With the introduction of higher-accuracy second-generation CCE (PillCam^®^COLON-2), the CCE’s usefulness for the noninvasive assessment of mucosal disease activity is currently being explored [16].

### 3.1. Ulcerative Colitis

In 2018, Hosoe et al. aimed to develop an endoscopic severity score for ulcerative colitis (UC) using the PillCam^®^COLON-2 [17]. Patients diagnosed with UC were enrolled prospectively and underwent colonoscopy and CCE on the same day. The CSUC score was developed using the collected CCE videos, which four blinded IBD experts scored. The descriptors validated with the Ulcerative Colitis Endoscopic Index of Severity (UCEIS) were used as the candidate items, some of which were automatically assessed using the workstation.

The UCEIS descriptors used in the CSUC score including vascular pattern (graded as normal, patchy obliteration and obliterated), bleeding (graded as none, mucosal, luminal mild, and luminal moderate or severe) and erosions and ulcers (graded as none, small erosions ≤ 5 mm, superficial ulcer ≥ 5 mm, and deep ulcer) [18] are shown in Table 3.

Each of the validated UCEIS descriptors was divided into proximal and distal parts at the splenic flexure and then individually assessed. This was thought to express the inflammation severity more accurately than conventional scoring, which takes into account the point of maximum intensity. The visual analogue scale (VAS) was simultaneously used by the image readers to score the inflammation of the entire colon between 0 “completely normal” and 100 “worst ever seen”.

In the meantime, the UCEIS of the proximal and distal parts was scored by two expert endoscopists using colonoscopy videos. This was considered as the gold standard.

The descriptors that contribute to the VAS were evaluated, and a model (estimating each item’s weighting) to predict the VAS was constructed. Moreover, the score’s contribution by colonic region (proximal, distal, or the sum of both) was accounted for. Through multiple regression analysis, the final scoring system was determined as “vascular pattern sum (proximal + distal) + bleeding sum + erosions and ulcers sum (minimum–maximum, 0–14)” and was named Capsule Scoring of Ulcerative Colitis (CSUC).

In terms of score validation, the authors showed an overall mean-weighted intra-observer agreement kappa value of 0.86 (95% CI 0.69–1.00). The interobserver kappa value was 0.52 (95% CI 0.45–0.59). This was low due to the lower dispersion of scores concerning bleeding. The proportion of agreement was, however, very high, with 0.96 (95% CI 0.94–0.98) for proximal and 0.90 (95% CI 0.87–0.93) for distal. The authors then compared the CSUC and UCEIS calculated from Pillcam^®^COLON-2 and OC videos, respectively. The correlation coefficients of these two endoscopic scores with a clinical activity score and faecal and blood biomarkers were calculated.

As a clinical score, the Lichtiger index was used. This score consists of eight clinical variables: number of bowel openings daily, nocturnal diarrhoea, percentage of visible blood in the stool, faecal incontinence, abdominal pain/cramping, general well-being, abdominal tenderness and need for anti-diarrhoeal agents. The score ranges from 0 to 21 points (remission is defined ≤3, mild activity 4–8, moderate 9–14, and ≥15 severe disease). The correlation coefficients of CSUC and UCEIS with the Lichtiger index were 0.60 and 0.48, respectively (95% CI 0.42–0.79).

In terms of blood biomarkers, the correlation coefficients of CSUC and UCEIS with WBC were found to be 0.40 and 0.33 (95% CI 0.31–0.48), respectively, and for CRP, these were 0.20 and 0.13 (95% CI 0.00–0.45), respectively.

A moderate association between CSUC and faecal calprotectin (FC) was present with a correlation coefficient of 0.46 (95% CI 0.19–0.72), whilst the correlation between UCEIS and FC was found to be 0.50. In 2015, Theede et al. demonstrated that FC can be used as a surrogate for mucosal healing as it is strongly associated with clinical disease activity (Partial Mayo Score), histological disease activity and endoscopic disease activity (UCEIS) [19]. Given the comparable correlations between FC, UCEIS and CSUC, there is a potential role for the use of Pillcam^®^COLON-2 in assessing mucosal healing.

CSUC may also be used as a predictor for the risk of relapse during clinical remission. In a retrospective observational study by Matsubayashi et al., patients were more likely to maintain clinical remission for a year if they had a CSUC score of ≤1 following 6 months of successful induction treatment. The calculated correlation coefficient for CSUC with FC was 0.50 (2-tailed *p*-value < 0.01) in this study. In addition, a CSUC ≥1 was shown to be a predictor of relapse through receiver operator characteristic curve analysis (area under the curve of 0.82, sensitivity of 83.3%, and specificity of 58.6%) [20].

Overall, the CSUC is a user-friendly score consisting of three simple parameters which can be easily generated using capsule endoscopy software; however, it will need further validation to determine whether Pillcam^®^COLON-2 can be used to accurately determine the UC inflammation extent [16].

### 3.2. Crohn’s Disease

The usefulness of CCE in Crohn’s disease (CD) remains unclear, contrasting with the established role of SB capsule endoscopy (SBCE) for the assessment of SB mucosal disease activity. In order to standardise the SB inflammatory burden in CD, two validated capsule endoscopy scores have been developed over the years, namely the Lewis score and the Capsule Endoscopy Crohn’s Disease Activity Index (CECDAI or Niv score). The CECDAI evaluates three endoscopic parameters: inflammation severity, disease extent and presence of strictures.

With the current availability of Pillcam^®^COLON-2, which allows for the visualisation of colonic inflammatory disease activity, Niv et al. extended the validated CECDAI score to include the colon, introducing the novel CECDAIic score, allowing for an objective panenteric assessment of CD inflammatory activity (Table 4).

A high level of inter-observant agreement was achieved for all parameters in the CECDAIic score, other than for strictures of the proximal small bowel and distal colon (C1 and C4). The panenteric Kendall coefficient was 0.77, whilst that for the small bowel was 0.85 [21].

This statistically significant high correlation was reproduced by Ariera et al. in a separate study, achieving a superior Kendall coefficient of 0.94 [22]. The authors also showed a high inter-observer concordance for all the subscores of CECDAIic (A1 = 0.91; B1 = 0.95; C1 = 1; A2 = 0.91; B2 = 0.91; C2 = 0.87; A3 = 0.84; B3 = 0.80; C3 = 1; A4 = 0.94; B4 = 0.88; C4 = 1; *p* < *0*.001). However, as this was a retrospective study with a small number of patients, further validation with prospective studies with a larger number of patients is required.

Furthermore, this study examined the correlation between CECDAIic, FC and CRP, respectively (correlation coefficients 0.82 and 0.52, respectively). Although a strong correlation between FC and CECDAIic was established, 66.7% (*n* = 2) of the three patients with normal FC were still found to have significant inflammatory activity on panenteric capsule endoscopy (Pan-CE). This is in line with previous evidence that biomarker levels such as FC tend to correlate modestly with small bowel disease activity.

Several potential advantages to using Pan-CE over OC followed by SBCE include patient preference given the need for just one bowel preparation and fewer days off work with Pan-CE. Parodi et al. showed that out of 177 patients, 40.7% (*n* = 72) would prefer capsule endoscopy over OC for future colorectal examinations—the latter was preferred in just 22.6% (*n* = 40). The rest (36.7%) expressed no preference in this study [23]. 

The comparison of findings on Pan-CE to OC and SBCE established a good correlation between Pan-CE and OC (r = 0.6667, *p* < 0.035) and Pan-CE and SBCE (r = 0.896, *p* < 0.0004) for colonic and SB disease activity, respectively [24].

The currently available data appear to support the use of Pan-CE and the CECDAIic score in order to objectively assess the inflammatory disease activity in SB and colonic CD. The CECDAIic is reproducible, with a high degree of inter-observer agreement for the overall score and also for its individual sub-scores. In the current treat-to-target era, Pan-CE and CECDAIic use may be superior to biomarkers such as FC in establishing mucosal healing [22].

Pan-CE may also be a more cost-effective strategy over OC and imaging in the monitoring of CD. In a model study by Saunders et al., the calculated annual savings were USD 1135 per patient exclusively monitored with Pan-CE. The authors argue that this is likely due to the earlier recognition of active disease and timely initiation of treatment, thus offsetting the costs incurred with the need for inpatient care and surgery [25]. The potential cost-savings between the use of biomarkers such as FC and Pan-CE remain to be investigated.

## 4. Polyp Detection and Size Estimation

The role of identifying and removing adenomatous polyps during colonoscopy in the prevention of colorectal cancer is well-established. Poor adherence to OC by the screening population is an important limiting factor of screening colonoscopy due to its invasive nature and associated embarrassment. It also has a role in the detection or surveillance of colonic polyp(s) if the previous OC was incomplete or in the presence of severe medical co-morbidities where sedation would be deemed a high risk [26].

Various studies have demonstrated that CCE could visualize colonic segments missed by colonoscopy in 84.0–93.5% of patients [27,28,29,30,31].

However, one small retrospective pilot study did not demonstrate any role of CCE on incomplete OC [30]. 

The second-generation colon capsule, Pillcam^®^COLON-2, has demonstrated a much better performance in detecting colonic polyps. The integrated polyp size estimation tool within the RAPID software (Given^®^Imaging Ltd., Yokneam, Israel) when using Pillcam^®^COLON-2 allows the ease and reliability of measurement of polyp size in millimetres.

In a meta-analysis in 2016, comparing Pillcam^®^COLON-2 with OC, CCE detected polyps > 10 mm with 87% sensitivity (95% CI, 81–91%) and 95.3% specificity (95% CI, 91.5–97.5%) and identified all 11 invasive cancers detected by colonoscopy (100% sensitivity/specificity) [32]. In addition, Hagel et al. compared Pillcam^®^COLON-2 with OC, using polyp-by-polyp findings with respect to location as defined by two highly experienced SB capsule endoscopists and specifically trained for CCE. The per-finding analysis for CCE polyp location estimation demonstrated sensitivity and specificity rates of 90.1% and 76.9%, respectively [33].

An additional advantage to Pillcam^®^COLON-2 over its first-generation predecessor is the ability to differentiate between adenomatous polyps and hyperplastic polyps with a sensitivity of 91.2% and specificity of 88.2% [34]. This is possible through digital imaging processing technologies such as flexible spectral imaging colour enhancement (FICE) and blue mode (BM). Therefore, for those patients in whom a hyperplastic polyp is identified on CCE, OC might not be necessary.

One of the main competitors for CCE is CTC. In a study in patients with positive immunohistochemical faecal occult blood tests (iFOBTs) who underwent CCE (using Pillcam^®^COLON-2), CTC, and colonoscopy, CCE and CTC displayed identical performances in CRC screening. CCE identified ≥6 mm polyps, with an 88.2% sensitivity and 87.8% specificity, whereas CTC had an 88.2% sensitivity and 84.8% specificity [35]. In a more recent study, CCE (Pillcam^®^COLON-2) was superior to CTC for the detection of polyps ≥ 6 mm and noninferior for the identification of polyps ≥ 10 mm [36].

There has been an increased demand for the use of noninvasive screening tests for CRC given the scarce resources associated with the COVID-19 pandemic. Stool-based DNA testing is one such modality which, when compared to OC as the reference standard in the DeeP-C cross-sectional study, was found to have sensitivity and specificity rates of 92.3% and 86.6%, respectively, for detecting CRC. Despite CCE having a comparable CRC detection of 93%, its potential drawback as a screening tool in CRC detection lies within the need for bowel preparation and variable completion rates [3,37].

## 5. Discussion

In an effort to still ensure timely diagnoses in the setting of curtailed nonemergency services as a result of the COVID-19 pandemic, the role of CCE as a safe, noninvasive method to investigate the colon is being increasingly recognised. CCE is being used in various aspects of endoscopic practice traditionally associated exclusively with OC such as CRC screening and mucosal disease activity assessment in IBD.

The use of OC as a CRC screening tool may be limited by the poor acceptance rate owing to patients’ concerns regarding its invasive nature, perceived lack of privacy, and potential adverse effects, together with logistical and financial implications from work-related absences [38]. CCE is an alternative investigation which could potentially mitigate such valid patient concerns, whilst still demonstrating comparable colonic polyp and CRC detection rates to OC as discussed previously [32]. Therefore, using CCE as a screening test for CRC could improve patient acceptance rates, whilst possibly reserving OC for therapeutic purposes. According to the European Society of Gastrointestinal Endoscopy (ESGE), CCE can be used in average-risk patients for CRC screening and as an adjunctive test in patients with a prior incomplete OC, or in whom the latter is contraindicated or refused [39]. 

Considering that the current estimated rate of incomplete OC is 1–43%, CCE lends itself as an important alternative to visualize the colonic mucosa in its entirety [40]. In a prospective study including 75 patients, CCE was able to progress beyond the colonic segment at which OC stopped in 91% of patients with a prior history of incomplete OC [30]. This resulted in the diagnosis of additional significant findings in 36% of same-day cases and in 48% of rescheduled ones. Moreover, a prospective, comparative trial between CCE and CTC in patients with incomplete OC demonstrated a relative sensitivity of CCE compared to CTC of 2.0 (95% CI 1.34 to 2.98) for polyps ≥ 6 mm. For larger polyps ≥ 10 mm, the relative sensitivity of CCE over CTC was 1.67 (95% CI 0.69–4.00). Positive predictive values for polyps ≥ 6 and ≥ 10 mm were 96% and 85.7%, and 83.3% and 100% for CCE and CTC, respectively [41]. In this study, both CCE and CTC achieved complete colonic evaluation in 98% of cases. This would suggest that CCE is a potentially more suitable investigation after an incomplete OC given its superior diagnostic yield over CTC [42]. 

Beyond the realm of incomplete OC, CCE is also suitable for patients considered too high-risk for OC or unwilling to undergo OC. From a prospective study of 70 high-risk patients who had refused or were unable to undergo OC, CCE revealed significant findings in 34% of patients [43].

In terms of cost-effectiveness, a computer model based on a Markov process in a hypothetical population of 100,000 individuals aged 50 and over undergoing a 10-yearly screening procedure demonstrated an incremental cost-effectiveness (compared with no screening) of OC and CCE of USD 16,165 and USD 29,244 per life-year saved, respectively. This showed that the OC program was less costly and more effective than CCE. The authors, however, showed that if the initial compliance to CCE were to be 30% better than OC, then CCE becomes the more cost-effective option [44].

The choice between CCE, OC and CTC can be challenging in everyday practice. The decision relies on several different factors such as patient consent, fitness for sedation, ability to tolerate bowel preparation, acceptable exposure to ionizing radiation, the need to assess for extra-colonic pathology, and also the availability and expertise for each and every modality. Table 5 summarizes pertinent variables to consider before choosing a suitable modality for a particular patient.

The various CCE scores discussed earlier are summarized in Table 6 and Table 7. These ensure good clinical practice, which can be audited as these allow for the standardisation of reporting between physicians, increase objectivity and enable continuity of care. The availability of these scores further supports the clinical use of CCE beyond CRC screening, but also in disease activity monitoring in IBD.

## 6. Conclusions

CCE is emerging as a safe, noninvasive and relatively painless procedure. In a recent meta-analysis, the pooled adverse event rate for capsule retention was 0.26% (95% CI 0.00–0.77), whilst the procedural discomfort rate was 0.81% (95% CI 0.15–1.80). The authors also demonstrated that the CCE technical failure rate was 1.76% (95% CI 0.76–3.06) [47]. This was defined as equipment malfunction, recording gaps, short battery life duration, activation failure and inability to download images.

Colonic incomplete examination, defined as CCE not excreted or not reaching the rectum within the recording time, was 19.19% (95% CI 14.06–25.88) [47]. 

Its application as a safe diagnostic tool, associated with comparable sensitivity and specificity rates to OC for the detection of colonic polyps, as well as its role in assessing mucosal disease activity in IBD, makes the second-generation CCE a suitable diagnostic endoscopic modality alternative to OC.

CCE is, therefore, a potentially beneficial method of investigating patients by reducing the burden on the reinstatement of curtailed endoscopy services in the post-COVID era. Nevertheless, with the average completion time for OC being 30–60 min, CCE is potentially more cumbersome considering the transit time and subsequent dedicated reading time averaging 25–65 min depending on reader experience [48].

## Figures and Tables

**Figure 1 jcm-10-02372-f001:**
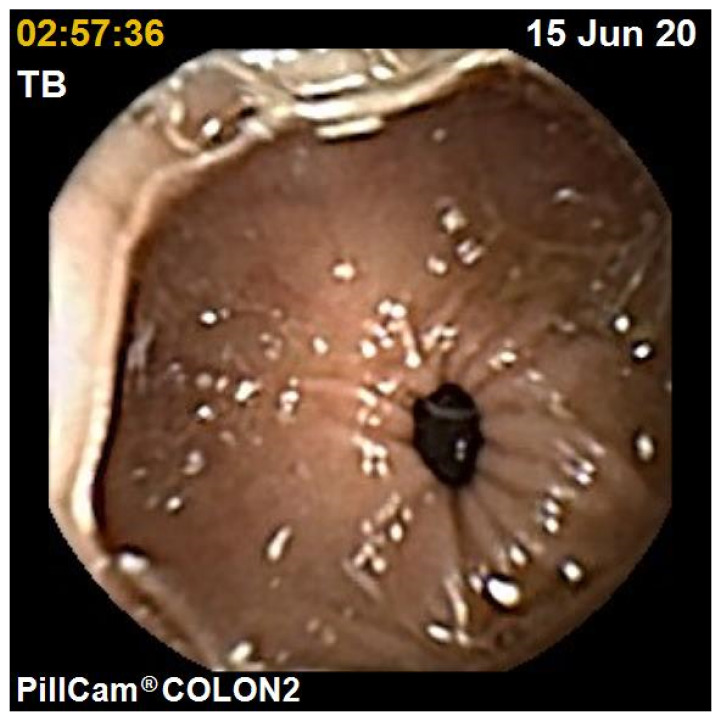
Normal colon.

**Figure 2 jcm-10-02372-f002:**
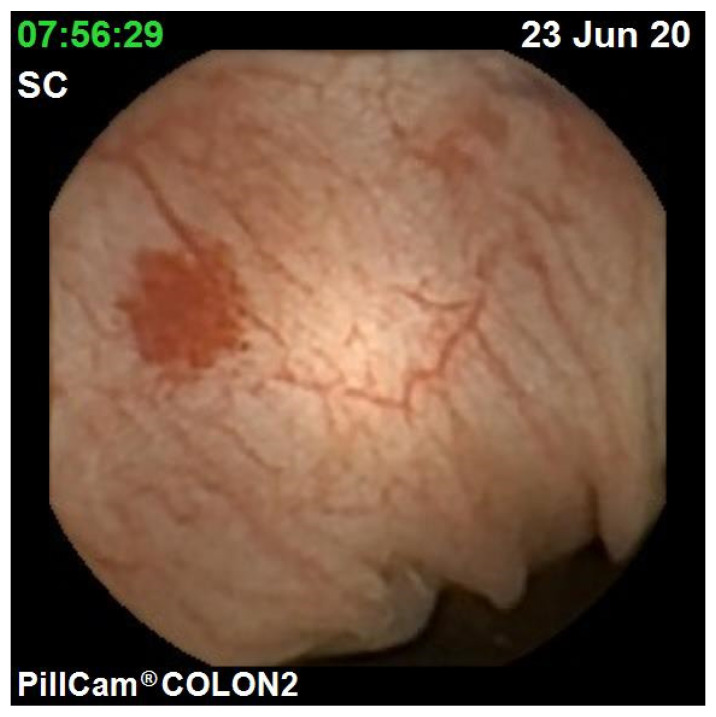
Colonic angiodysplasia.

**Figure 3 jcm-10-02372-f003:**
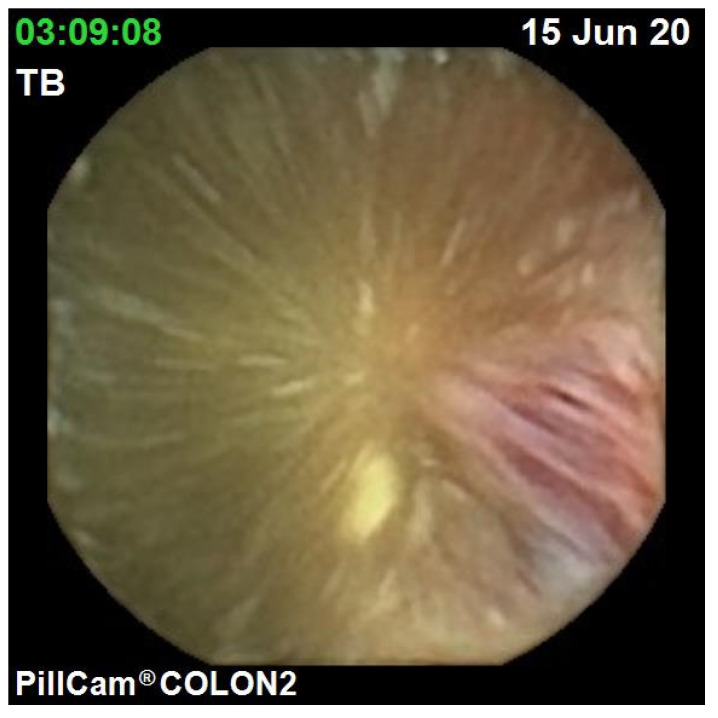
Haemorrhoid.

**Figure 4 jcm-10-02372-f004:**
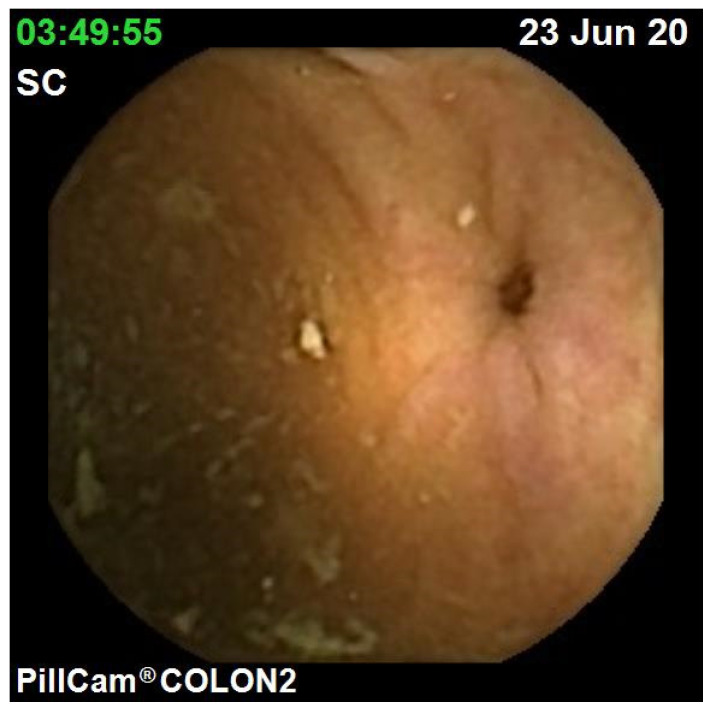
Colonic diverticulum.

**Table 1 jcm-10-02372-t001:** Grading of bowel cleanliness.

Grade	Colon Cleanliness
A	Large amount of faecal residue
B	Enough faeces or dark fluid present to preclude a completely reliable examination
C	Small amount of faeces or dark fluid, but not enough to interfere with examination
D	No more than small bits of adherent faeces

**Table 2 jcm-10-02372-t002:** Leighton–Rex Scale.

Rating	Description
Cleansing Level Scale	
Poor	Inadequate. Large amount of faecal residue precludes a complete examination
Fair	Inadequate but examination completed. Enough faeces or turbid fluid to prevent a reliable examination
Good	Adequate. Small amount of faeces or turbid fluid not interfering with examination
Excellent	Adequate. No more than small bits of adherent faeces
Bubbles Effect Scale	
Significant	Bubbles that interfere with the examination. More than 10% of surface area obscured by bubbles
Insignificant	No bubbles or bubbles that do not interfere with the examination. Less than 10% of surface area obscured by bubbles

**Table 3 jcm-10-02372-t003:** CSUS descriptors and definitions.

Descriptor	Likert Scale Anchor Points	Definition
Vascular Pattern	Normal (0)	Normal vascular pattern
	Patchy obliteration (1)	Obliterated area ≤ 30%
	Obliterated (2)	Obliterated area > 30%
Bleeding	None (0)	No visible blood detected by image reading software
	Mild (1)	No bleeding picture detected by image reading software ≤ 10
	Severe (2)	No bleeding picture detected by image reading software > 10
Erosions and ulcers	None (0)	Normal mucosa, no visible erosions or ulcers
	Erosions (1)	Tiny (≤5 mm) defects in the mucosa
	Superficial ulcer (2)	Larger (>5 mm) defects in the mucosa
	Deep ulcer (3)	Larger (>5 mm) and deeper excavated defects in the mucosa, with a slightly raised edge

**Table 4 jcm-10-02372-t004:** CECDAIic score ^1^.

Parameter	
A	Inflammation score
0	None
1	Mild to moderate oedema/hyperaemia/denudation
2	Severe oedema/hyperaemia/denudation
3	Bleeding, exudate, aphthous ulcer, erosion and small ulcers (<0.5 cm)
4	Moderate ulcer (0.5–2 cm), pseudopolyp
5	Large ulcer (>2 cm)
B	Extent of disease score
0	None
1	Focal disease (single segment)
2	Patchy disease (multiple segments)
3	Diffuse disease
C	Narrowing (stricture)
0	None
1	Single passed
2	Multiple passed
3	Obstruction

^1^ Segmental score = A × B + C. Total score = (A1 × B1 + C1) + (A2 × B2 + C2) + (A3 × B3 + C3) + (A4 × B4 + C4). (1) Proximal small bowel; (2) Distal small bowel; (3) Right colon; (4) Left colon.

**Table 5 jcm-10-02372-t005:** Comparison between CCE, OC and CTC.

	CCE	Diagnostic OC	CTC
Bowel preparation	PEG + booster + prokinetic	PEG	Laxative-free regimens using faecal tagging solution [11]
Ionising radiation exposure	None	None	≤3–6 mSv [45]
Need for sedation	None	Yes	None
Suitability in high-risk patients	Potentially unsuitable given relative aggressiveness of bowel preparation	Potentially unsuitable given sedation-related risks	Suitable owing to better tolerated bowel preparation and no need for sedation
Bowel insufflation	None	Yes	Yes
Extra-colonic findings	None	None	Yes
Detection of polyps ≥ 6 mm (sensitivity, specificity, respectively)	88.2%, 87.8% [35]	‘Gold standard’	88.2%, 87.8% [35]
Average cost	USD 950 [46]	USD 877 [44]	USD 500 [42]

**Table 6 jcm-10-02372-t006:** Summary of bowel cleanliness scores.

Type of Computation	Name of Score	Quantitative vs. Qualitative	Colonic Segmental Division	Interobserver Agreement
Operator-Dependant	Leighton–Rex	Qualitative	5 segments: caecum, right colon, transverse, left colon, and rectum	Kendall’s coefficient: 0.806 [14]
	CC-Clear	Quantitative	3 segments: right colon, transverse, and left colon	Kendall’s coefficient: 0.911 [14]
Computer-Dependant	CAC	Qualitative	Pancolonic	Pearson’s: 0.53 [15]

**Table 7 jcm-10-02372-t007:** Summary of CCE scores used in IBD.

Type of IBD	Name of Score	Quantitative vs. Qualitative	Parameters of Intestinal Pathology Used	Bowel Segmental Division	Scoring Formula	Interobserver Agreement
UC	CSUC	Quantitative	Uses UCEIS descriptors: Vascular pattern; Bleeding; Erosions and ulcer.	Proximal colon; Distal colon (divided at the splenic flexure)	Vascular pattern sum (proximal + distal) + bleeding sum + erosions and ulcers sum	Mean-weighted kappa value: 0.52 (95% CI 0.45–0.59) [17]
CD	CECDAIic	Quantitative	Inflammation (A); Extent of disease (B); Presence of strictures (C)	Proximal small bowel (1); Distal small bowel (2); Right colon (3); Left colon (4)	Total score = (A1 × B1 + C1) + (A2 × B2 + C2) + (A3 × B3 + C3) + (A4 × B4 + C4)	Kendall coefficient: 0.77 [21]

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
