# Peer review of "Scoring Systems for Clinical Colon Capsule Endoscopy—All You Need to Know"

_jcm, 2021, doi:10.3390/jcm10112372_

Round 1

Reviewer 1 Report

  • Define the abbreviation the first time they appear (for example, COVID-19)

  • Explain better this concept: “… this also resulted in a renewed interest in the use of colon capsule endoscopy (CCE) for minimally invasive colonic exploration”

  • Comparing CCE with CTC, CCE requires a heavy bowel preparation, that is unsuitable for some patients categories (for example kidney or hepatic failure): discuss this aspect

  • Use oxford comma

  • “CCE plays a major role in determining mucosal disease activity in inflammatory bowel disease (IBD), this varying from response to treatment to mucosal healing and surveillance.”

This is not true. CCE is not currently used in many IBD centres due to the impossibility of evaluating the histological data and the contraindication of stenosing in Crohn's disease patients.

  • “the current treat-to-target era, Pan-CE and CECDAIic use may be superior to biomarkers such as FC in establishing mucosal healing”

What about the cost?

  • “CCE is therefore a potentially beneficial method of investigating patients by reducing the burden on the reinstatement of curtailed endoscopy services in the post-COVID era.”

Compare the time needed to complete an OC or a CCE examination

Reviewer 2 Report

  • Table numbering is not consistent. For example, the table caption has ‘Table A1’, but the main text has ‘Table 1’.
  • Figure numbering is not consistent. For example, the figure caption has ‘Figure A1’, but the main text has ‘Figure 1’.
  • It is better to provide some comparison between CCE and Cologuard for CRC detection.
  • There is little novelty of the paper. It looks that it is a summary of a number of related papers.
  • Basically, there is little (or no) difference between BBPS and Table 1. It is better to explain their difference(s).
  • Page 02, Line 59: "Validated scores for conventional colonoscopy cannot be used for colon CCE"- It is better to explain why.
  • Page 05, Line 149: "Furthermore, still frames and not an entire video was made use of...." – Its English is not clear.
  • Page 05, Line 140 ‘Computer-Dependemtnt Scores’ needs to be ‘Computer-Dependent Scores’.

Reviewer 3 Report

We have reviewed the paper entitled “Scoring Systems for Colon Capsule Endoscopy – all you need to know!”

This review article reports on most of the available scores evaluating the quality, sensitivity, and specificity of colonic capsule endoscopy (CCE) in colon exploration.

By this presentation, the reader has an extensive understanding of all the scores.

The main weakness of this review is the lack of interpretation of these scores relative to the objective of the CCE exploration. In fact, there is no suggestion for the way to use these scores. For example, the main interest of this paper could be to suggest to the practitioner evaluating the exploration results a way to present those results.

There is one score for each potential disease and such a score cannot be adapted to complex cases including multiple investigations. For example, in the case of a patient with Crohn’s disease, an exploration for polyps and a bleeding occurrence would mix many scores together.

A summary table presenting separately the cleansing scores, operator and computer-dependent scores and disease-related scores (inflammatory bowel disease, Crohn’s disease, and polyp detection) could be interesting by proposing a theoretical CCE exploration report useful for any operator.

Reviewer 4 Report

This is an interesting review article summarising the various scoring systems in CCE. The manuscript is well written and well structured. I only have some minor comments:

  1. I would suggest that the authors add a small paragraph with the limitations of these scoring systems that may restrict their use in clinical practice - for example, complexity of some of these scoring systems, need for further validation etc.
  2. minor spelling/language errors (eg. see line 140: correct to "computer-dependant scores" & lines 271-272 etc)
  3. In the section on UC and CSUC scoring system (section 3.1), I would add a supplementary table with the descriptors and scoring system, so that readers do not have to refer to the original publication to see the precise details of the scoring system
  4. In the section on polyp characterisation (section 4), I would include the recent reference on the use of colour differences on CCE to determine the type of the polyp (hyperplastic vs adenomas). 
    The Differential Diagnosis of Colorectal Polyps Using Colon Capsule Endoscopy. 
    Intern Med 2021. PMID: 33456043

Round 2

Reviewer 2 Report

The new version did not address the following two main concerns:

  • It is better to provide some comparison between CCE and Cologuard for CRC detection.
  • There is little novelty of the paper. It looks that it is a summary of a number of related papers.